# Resistance to Immune Checkpoint Inhibitors Secondary to Myeloid-Derived Suppressor Cells: A New Therapeutic Targeting of Haematological Malignancies

**DOI:** 10.3390/jcm10091919

**Published:** 2021-04-28

**Authors:** Alejandro Olivares-Hernández, Luis Figuero-Pérez, Eduardo Terán-Brage, Álvaro López-Gutiérrez, Álvaro Tamayo Velasco, Rogelio González Sarmiento, Juan Jesús Cruz-Hernández, José Pablo Miramontes-González

**Affiliations:** 1Department of Medical Oncology, University Hospital of Salamanca, 37007 Salamanca, Spain; figuero44@gmail.com (L.F.-P.); eduardo.teran.brage@gmail.com (E.T.-B.); varolopez94@hotmail.com (Á.L.-G.); jjcruz@usal.es (J.J.C.-H.); 2Institute for Biomedical Research of Salamanca (IBSAL), 37007 Salamanca, Spain; gonzalez@usal.es; 3Department of Haematology, University Hospital of Valladolid, 47003 Valladolid, Spain; atamayov@saludcastillayleon.es; 4Department of Medicine, University of Salamanca, 37007 Salamanca, Spain; 5Department of Internal Medicine, University Hospital Rio Hortega, 47012 Valladolid, Spain; 6Department of Medicine, University of Valladolid, 45005 Valladolid, Spain

**Keywords:** myeloid-derived suppressor cells (MDSCs), immune checkpoint inhibitors (ICIs), haematological malignancies, immune resistance

## Abstract

Myeloid-derived suppressor cells (MDSCs) are a set of immature myeloid lineage cells that include macrophages, granulocytes, and dendritic cell precursors. This subpopulation has been described in relation to the tumour processes at different levels, including resistance to immunotherapy, such as immune checkpoint inhibitors (ICIs). Currently, multiple studies at the preclinical and clinical levels seek to use this cell population for the treatment of different haematological neoplasms, together with ICIs. This review addresses the different points in ongoing studies of MDSCs and ICIs in haematological malignancies and their future significance in routine clinical practice.

## 1. Introduction

Myeloid-derived suppressor cells (MDSCs) are a heterogeneous population of immature myeloid lineage cells that include macrophages, granulocytes, and dendritic cell precursors [1,2]. This cell population has the peculiar ability to suppress both innate and adaptive immune activity [3,4]. The role of these cells is their inhibition of immune cells, mainly T cells, and, to a lesser extent, B and NK cells. In virtually all studies that have been carried out, it has been observed that these cells are associated with a worse response to cancer treatments and lower survival rates in patients with solid and haematological tumours [5,6]. Although most of the studies concerning MDSCs have been carried out in solid tumours, in recent years, the relationship of these cells with haematological malignancies and immune-mediated cytopenias has become more evident [7,8].

This group of cells has also been shown to promote the progression and formation of metastases through remodelling of the tumour microenvironment and angiogenesis through vascular endothelial growth factor (VEGF), fibroblast growth factor (FGF), and matrix metalloprotease 9 [9,10]. In this way, the presence of MDSCs in the tumour bed is not only a potential biomarker of the aggressiveness, response, and survival of the different tumours (both solid and haematological), it is also a probable therapeutic target in combination with the system’s immune checkpoint inhibitors (ICIs) [11,12]. The appearance of ICIs has also marked a milestone in the treatment of different neoplasms, both solid and haematological. Several studies have evaluated the combination of agents directed against MDSCs and immunotherapy as a possible new treatment for different neoplasms. In this study, we review the role of these new investigational drugs in haematologic malignancies.

## 2. Definition and Role of MDSCs

In the last 5–10 years, a better characterization of MDSCs has been observed in various studies. Likewise, it has been seen that these myeloid cell populations show synergy with different regulatory mechanisms of the immune system, which may be essential in the treatment of different neoplasms. MDSCs are composed of a heterogeneous population of immature myeloid cells (IMCs) in various states of transcriptional activity and differentiation [13]. The myeloid lineage expands clonally under pathological conditions, where increased production of myeloid leukocytes in the bone marrow carries out an important and fundamental defence against bacteria, tumours, or other external agents [14,15].

At the time of induction of the tumour microenvironment (TME), a dysregulation of the immune system occurs that leads to various alterations in the tumour, preventing immune system action [16]. Along with the dysregulation of the immune system, there is also an increase in the expression of inflammatory cytokines. The combinatorial effects of these cytokines can alter myeloid cell differentiation with increased MDSC production, creating an IMC spectrum that is morphologically analogous to granulocytes and monocytes. Through a continuous process of chronic inflammation, different haematologic tumours are also capable of amplifying myelopoiesis, which contributes to the progression and spread of the tumour [17,18]. MDSCs are attracted to the tumour and its microenvironment through the secretion of different chemotactic substances. Among the different mechanisms that contribute to this process are immunoevasion through the induction of anergy of NK cells [19] and T cells [20], as well as the induction and facilitation of TME processes to promote tumour growth, create and establish a metastatic niche for the spread of cancer, promote angiogenesis, or improve tumour cell survival through its immunosuppressive activity [21]. From the above, it can be understood that MDSCs actively contribute to the ineffectiveness of the immune system in tumour control and therefore impede the efficacy of immunotherapy against cancer.

## 3. Classification of MDSCs

MDSCs can be divided into two groups: polymorphonuclear (PMN-MDSC) and monocytic (M-MDSC) [22,23]. In humans, MDSCs are identified by myeloid cell markers CD11b +, CD33 +, HLA-DR low/–, and lineage-specific antigen Lin-negative. In the case of M-MDSC, the target expression is CD11b + CD33 + HLA-DR- / CD14 + CD15-, and for PMN-MDSC, it is CD11b + CD33 + HLA-DR-/CD14-CD15 + [24,25,26]. Despite this classification, in the different tumours, both solid and haematological, it is possible to find the different populations of MDSCs together [27,28]. The different expression markers in the MDSCs are summarized in Table 1 [29,30,31]. M-MDSC and PMN-MDSC have the same expression of CD11b, CD38, CD39, CD40, CD45, CD62L, CD86, CD120, CD162, and PD-L1. It is important to bear in mind that the expression of the different MDSC detection biomarkers may vary depending on the tumour under study.

Despite the different existing efforts, there is no standardized method that allows the determination of a series of markers to study in the different populations of MDSCs [32], even though, in the future, it will be essential to characterize them in order to bring their in vitro applications to routine clinical practice. The differential expression of MDSC population markers in clinical practice is summarized in Table 2. A number of commonly expressed markers exist between M-MDSCs and PMN-MDSCs: CD13 (low in M-MDSC and high in PMN-MDSC), CD16 (high in M-MDSC and low in PMN-MDSC), CD33 (high in M-MDSC and low in PMN-MDSC), CD34 (high in M-MDSC and low in PMN-MDSC), CD11b, CD38, CD39, CD45, CD62L, CD73, CD115, and CD124.

## 4. MDSCs and Haematological Malignancies

In recent years, a multitude of studies have emerged that evaluate the role of MDSCs in haematological malignancies. Most of these studies have associated MDSCs with tumours in more advanced stages, a high tumour burden, and poor treatment results. However, the role of MDSCs must be analysed according to the different haematological neoplasms. In particular, current findings indicate that MDSCs can be considered prognostic markers in haematologic malignancies.

### 4.1. Lymphoma

The term lymphoma includes both non-Hodgkin’s lymphomas (NHL) and Hodgkin’s lymphoma (HL). They are clonal diseases of B, T, or NK cells in various stages of differentiation. These cells proliferate in lymph nodes and other lymphoid organs [7,8]. Although treatment strategies have been shown to be effective, a large number of patients may have a relapses of the disease, so new treatment options are being investigated, including therapies that act on the host’s immunosuppressive tumour microenvironment (TME) and immune cells, as is the case with checkpoint inhibitors. Patients have been shown to be affected by immunosuppressive cells, such as regulatory T cells (Tregs) and MDSCs, which can counteract the efficacy of these new therapies [14].

#### 4.1.1. MDSCs in Hodgkin’s Lymphoma

HL cells are surrounded by a TME, which produces the suppression of the immune response. In a work by Romano et al., subsets of undifferentiated M-MDSC, G-MDSC, and CD34 + MDSC were higher in peripheral blood samples from 60 patients with HL at diagnosis compared to healthy patients. It was also observed that the patients who presented better responses to chemotherapy had lower levels of CD4+ MDSCs. Likewise, CD34+ MDSCs were presented as a promising biomarker for the outcome in HL with a specificity and sensitivity of 92% and 89%, respectively [33].

In another study [34,35], higher levels of CD66b +/CD33dim/HLA-DR-G-MDSC were found in patients with B cell lymphoma at diagnosis, in contrast with healthy controls. A decrease of CD66b + MDSCs from patients’ peripheral blood mononuclear cells (PBMCs) increased the levels of proliferating T cells, showing that these MDSCs are immunosuppressive. Poor prognosis and decreased progression-free survival (PFS) were observed in patients with high MDSCs levels.

In one study, three subtypes of MDSCs were reported to be increased in newly diagnosed advanced-stage HL patients, and decreased after at least two cycles of chemotherapy with Adriamycin, bleomycin, vinblastine, and dacarbazine (ABVD) [36]. However, the immunosuppressive effect and the correlation with the outcome appeared stronger for G-MDSC than for M-MDSC in HL, and only G-MDSC increased in HL patients compared to healthy controls.

#### 4.1.2. MDSCs in Non-Hodgkin’s Lymphoma

Lin et al. [37] studied mononuclear cells in 40 patients with B cell NHL and reported that monocytes with a CD14 + HLA-DR low/− profile reduce host immune responsiveness by reducing IFNγ production and suppressing T cell proliferation. According to these reports, NHL patients with a higher number of CD14 + HLA-DR low/− cells had a more advanced stage of the disease.

Analysing various studies of patients with B and T cell NHL, M-MDSCs were found to have higher levels in peripheral blood compared to healthy donors; these elevated levels were correlated with an advanced stage, higher recurrence, higher International Prognostic Index (IPI) score, and lower PFS. M-MDSCs may return to normal after patients achieve remission. Removal of MDSC from patients could re-establish T cell proliferation [38,39,40,41]. G-MDSCs have also been reported to accumulate in patients with HL and B-NHL compared to healthy controls, while reduction of CD66b + cells may re-establish T cell proliferation, similarly to the decrease in M-MDSC [35]. Elevated levels of MDSC, especially M-MDSC, were found in extranodal natural killer (NK)/T cell lymphoma (ENKL) at diagnosis, and these levels were predictors of overall survival. Furthermore, IL-17, ARG1, and iNOS expressions were elevated in ENKL patients, and inhibition of iNOS and ARG1 restored T cell proliferation [42].

In a study applying adoptive NK cell infusion therapy for NHL patients, Bachanova et al. showed that high levels of MDSC are associated with a lack of clinical response, in line with preclinical results that showed that MDSCs mediate NK cell inhibition. In this phase II clinical trial of patients with relapsed or refractory NHL [43], baseline levels of MDSC in peripheral blood were associated with a positive response to haploidentical donor NK cell therapy combined with rituximab (anti-CD20). Interestingly, the non-responders had elevated levels of the T cell immunoreceptor, with immunoglobulin and ITIM (immunoreceptor tyrosine-based inhibition motif) (TIGIT) domains in their T cells. This is consistent with previous work indicating that low TIGIT levels in NK cells confer resistance to MDSC-mediated suppression, and suggests that therapeutic efficacy can be further enhanced by blocking MDSC signalling [44]. In summary, MDSCs, especially M-MDSCs, may participate in carcinogenesis by restraining T cell proliferation in lymphoma.

### 4.2. Multiple Myeloma

Monoclonal gammopathies are clonal proliferations of B cells in the last maturing stages (plasma cells). They include multiple myeloma (MM), which is a proliferation of neoplastic plasma cells in the bone marrow that are usually associated with typical symptoms (anaemia, bone lesions, kidney failure, hypercalcemia). MM patients usually present increased levels of immunosuppressive cells and cytokines. In one study, MDSCs levels were increased in patients with MM compared to healthy controls [45]. It was subsequently shown that the levels of M-MDSCs were positively correlated with recurrent MM and negatively with response to treatment [46]. Likewise, it is suggested that G-MDSCs could play a key role in the pathogenesis of MM. G-MDSCs have also been reported to accumulate in both the BM and PB of MM patients compared to healthy controls, leading to higher MM activity and lower PFS [47,48,49,50].

Arg-1, iNOS, ROS, and TNF-α were found to be overexpressed by MDSCs [50]. In a recent report, PMN-MDSCs and their function through increasing Arg-1 are associated with the progression of MM. PMN-MDSC and arginase are raised in MM, and are potential biomarkers of a poor treatment response [51].

Wang et al. [46] and Favarolo et al. [48] studied the immunosuppressive capacities of MDSCs in other immune cells, such as Tregs. Increased levels of Tregs have been found in MM patients compared to controls, as has the induction of Tregs by MDSCs in a cell contact-dependent manner. There are conflicting reports on the effect of the proteasome inhibitor bortezomib and the immunomodulatory agent lenalidomide on MDSCs in the treatment of MM. In various studies [46,52], treatment with bortezomib combined with other drugs, such as dexamethasone or lenalidomide, reduced MDSCs levels in PB.

G-MDSCs can regulate angiogenesis in MM through the expression of P-element Induced Wimpy testis (PIWI)-interacting RNA (piRNA)-823, which promotes DNA methylation; G-MDSCs also increase the carcinogenic potential of MM cells in vitro and in vivo [52,53,54]. It was recently reported that treatment with the demethylating agent decitabine (DAC) inhibited MM cell proliferation in Merwin-11 plasma cell tumour cells (MPC11) and enhanced the immune response of autologous T cells by depleting M-MDSCs. The study demonstrated that MDSC depletion by DAC could decrease MM proliferation, considering that MDSCs are fundamental for MM progression [55]. In their study, Nakamura et al. [56] suggested that IL-18 acts as a key factor for immunosuppression in the MM BM niche through the generation of MDSCs

### 4.3. Leukaemia

In contrast to the robust body of research addressing lymphoma and MM, studies on MDSC in leukaemia have been relatively limited.

#### 4.3.1. Acute Leukaemias

Several studies reported that MDSCs were accumulated in the PB and BM of patients with acute myeloid leukaemia (AML) in contrast with healthy controls [57], and the therapeutic response of patients with B cell acute lymphoblastic leukaemia (ALL-B) was positively correlated with elevated levels of G-MDSC in both PB and BM [58]. A recent study demonstrated that the V-domain immunoglobulin suppressor of T cell activation (VISTA) immune checkpoint protein is highly expressed in MDSCs in the PB of AML patients. VISTA expression is associated with T cell immunosuppression. Regulation of VISTA by siRNA reduced the ability of MDSCs to inhibit CD8 + T cell activity. Furthermore, a strong positive association was observed between VISTA expression on MDSCs and PD-1 expression on T cells in AML [59].

Using multiplex immunohistochemistry, Hotari et al. [60] found that M1-like macrophages, granzyme B + CD57 + CD8 + T cells, and CD27 + T cells decreased in BM biopsy samples from 52 ALL patients compared to 14 healthy controls, whereas M2-like macrophages and MDSCs increased. Increases in MDSCs and immune markers, such as PD-1 and CTLA-4, have also been associated with immune regulation in ALL [60]. IL-13 secreted by ILC2s in patients with acute promyelocytic leukaemia (APL) increased M-MDSCs levels and amplified tumour development. ILC2-MDSC secretion may be diminished by ATRA treatment [61].

#### 4.3.2. Myeloproliferative Neoplasms

The increasing recognition of MDSCs as markers of a poor prognosis has recently been extended to chronic myeloid leukaemia (CML). Imatinib and dasatinib, which disrupt BCR- and ABL-mediated oncogenic signalling in CML, depleted the levels of MDSCs and their biomarkers IL-10, ARG1, and myeloperoxidase [62]. Furthermore, in high-risk CML patients, levels of ARG1-expressing PD-L1 + MDSCs increased, as did PD-1 expression on T cells, and MDSC levels decreased to normal after therapy with imatinib [63]. In a recent study, TKI therapy decreased the percentage of G-MDSC, but only dasatinib-treated patients experienced a significant reduction in the number of M-MDSCs. Therefore, M-MDSCs were identified as a prognostic factor for dasatinib-treated patients [64]. According to the current literature, MDSC levels in CML are reduced after treatment with TKI and IFN-α interferon in a time- and dose-dependent manner. A reduction in MDSC levels was observed after short-term IFN-α treatment, but MDSC levels increased with long-term therapy. In fact, chronic exposure to low doses of IFN-α may induce a suppressive TME through the activation of MDSCs [65,66,67,68]. This study showed that the addition of IFN-α to TKI therapy boosts a suppressive TME, with higher levels of Treg, MDSC, and CD4 + PD1 + T cells.

#### 4.3.3. Chronic Lymphocytic Leukaemia (CLL)

In patients with CLL, the levels of M-MDSCs were significantly increased at the time of diagnosis, suppressing T cell activation in vitro and inducing the formation of suppressor regulatory T cells [67]. A separate study included 49 patients with CLL, all of whom had increased levels of M-MDSCs and suppressed CD4 + T cell immune response, which correlates with a worse prognosis [68]. Furthermore, increased levels of M-MDSCs were observed in PB in 50 newly diagnosed CLL patients. Higher levels of M-MDSCs predicted poorer survival rates [69]. In another study, high levels of M-MDSCs predicted failure of chimeric antigen receptor T cell therapy in CLL [70].

### 4.4. Myelodisplastic Syndromes

Myelodysplastic syndromes (MDSs) are characterized by ineffective dysplastic haematopoiesis associated with aberrant expansion and activation of MDSCs within the bone marrow niche. Increased levels of MDSCs are correlated with a high risk of disease progression and a poor prognosis [65]. In one study, increased expression of PD-1 in haematopoietic stem and progenitor cells and PD-L1 in MDSCs was observed in MDS patients versus healthy donors. High concentrations of S100A9 produced by MDSC in the bone marrow niche of patients with MDS, together with IL-10, TGFβ, and activation of the myelocytomatosis proto-oncogene (MYC), induces PD-L1 formation to facilitate immune evasion [68]. The therapeutic effectiveness of these new agents directed against MDSCs in myelodysplastic syndromes is currently under investigation.

## 5. Immunotherapy for Haematological Malignancies

In animal models and in vitro studies, drugs directed against MDSC subpopulations have shown greater activity when combined with ICIs. This has opened a path in the investigation of ICIs in combination with other drugs (like anti-MDSC drugs). The following paragraphs address these investigations and the pathophysiology of ICIs in the treatment of haematologic malignancies in combination with anti-MDSC drugs.

TME is essential in tumour growth, development, and invasion, as well as in the interactions of the tumour with the immune system. [71,72]. Immune tolerance, which is a critical mechanism for immune invasion of cancer, remains an important barrier to effective antitumour therapy [73]. Although immunotherapy has shown effective activity in solid tumour therapies, its effects in haematologic malignancies remain inconsistent [74]. With some of the characteristics of different haematologic tumours, it is known that immunotherapy can provide unique opportunities, while also posing significant challenges.

Several characteristics of haematologic malignancies deserve special consideration regarding the use of immunotherapy in their treatment [75]. First, many such neoplasms involve lymphoid organs, which means that the very origin of both the neoplasm and the immunotherapy treatment is the immune system itself [76]. For example, neoplasms such as CLL or ALL are B cell cancers. In contrast to solid neoplasms, in haematological neoplasms, there is continuous contact with the immune system. Second, there may be an alteration in the normal development of the cells of the immune system due to the interaction of tumour cells with the immune cells and the invasion of lymphoid structures. Third, the interaction between tumour cells and the immune system in haematological neoplasms can lead to a series of growth stimuli that do not exist in solid neoplasms [77,78]. Finally, it is also important to note that many haematologic malignancies have a low mutation load and are therefore less immunogenic than many solid tumours [79].

Despite these differences, haematological neoplasms represent a challenge in the development of immunotherapy treatments based on ICIs. [80,81,82]. The blocking of the different checkpoints of the immune system earned the researchers Tasuku Honjo and James Allison the 2018 Nobel Prize in Physiology or Medicine [83,84]. This strategy has been shown to be effective in a large number of tumours, as evidenced by the success of CTLA-4 (cytotoxic T-lymphocyte antigen 4) and PD-1 (programmed cell death protein 1) pathway-blocking antibodies [85]. These molecules are negative regulators of the immune system, and these pathways are key to maintaining the antitumour function of the immune system, with tumour cells taking advantage of these mechanisms to evade immunological surveillance [86,87]. The regulation of these receptors by different drugs makes it possible to revitalize T cells and provoke tumour regression [88,89].

Among the ICIs, the drugs that have shown the greatest effectiveness in the treatment of haematological neoplasms are those that regulate the PD-1 pathway [90]. The PD-1 receptor is a type-1 transmembrane immunoreceptor expressed by activated T cells, NK cells, and B cells [91,92]. To prevent immune-related damage after T cell receptor (TCR) binding to the major histocompatibility complex (MHC) peptide, T cell activation is inhibited by the binding of PD-1 on T cells to its ligands PD-L1 (programmed death ligand 1) and/or PD-L2 (programmed death ligand 2) [93]. PD-1 ligands are known to be expressed in the tumour microenvironment by both tumour cells and non-tumour immune and stromal cells [94,95]. Therefore, the development of drugs that are capable of negatively regulating these pathways has permitted the development of antitumour therapies. Of the multiple drugs that have been developed, the two that have shown the greatest efficacy in haematological neoplasm therapies are pembrolizumab (Keytruda^®^, Merck, Kenilworth, NJ, USA) [96] and nivolumab (Opdivo^®^, Bristol-Myers Squibb, New York, NY, USA) [97].

### 5.1. Hodgkin’s Lymphoma

The first trials with PD-1 pathway-blocking drugs highlighted their effectiveness in certain lymphomas, especially classical Hodgkin’s lymphoma (cHL), which harbour an intrinsic susceptibility to PD-1 blockade [98,99]. CHL is defined by the presence of Hodgkin and Reed/Sternberg (HRS) cells within a robust inflammatory infiltrate, the treatment of which is based on polychemotherapy regimens with monoclonal antibodies [100]. However, a percentage of patients relapse or are refractory to existing therapies. In these cases, an almost universal overexpression of PD-L1 was observed in HRS cells, probably driven by 9p24.1 amplification, which led to the clinical investigation of PD-1 inhibitors in these tumours [101,102].

Knowing the above, significant activity was observed in phase I clinical trials of the drugs nivolumab (humanized IgG1 kappa monoclonal antibody) and pembrolizumab (humanized IgG4 kappa monoclonal antibody), which block the interaction of PD-1 with its PD-L1 and PD-L2 ligands [103]. The response to these drugs is clearly related to the degree of amplification of 9p24.1 [104]. In the CheckMate 039 (nivolumab) study [105], patients with relapsed or refractory cHL (R/R) had an overall response rate (ORR) of 87%, with 17% achieving a complete response (CR) and 70% achieved a partial response (PR). Among the 31 cHL patients who relapsed after brentuximab vedotin (BV) and autologous stem cell transplantation or who were ineligible for BV, the ORR was 58%, the CR rate was 19%, and the PR rate was 12%, with a median 24.9-month follow-up [106].

The phase II study with pembrolizumab KEYNOTE-087 evaluated 210 patients from three cohorts, in which the overall response rate was 69% and the overall CR rate was 22.4%. The ORR was 73.9% in patients with R/R cHL and with progressive lymphoma after autologous transplantation and BV, 64.2% in patients with chemo-resistant disease not eligible for transplantation, and 70.0% in patients without BV with relapse after transplantation [107]. These results led to both drugs being approved by the US Food and Drug Administration (FDA) and the European Medicines Agency (EMA) for the treatment of HL R/R.

Various studies have evaluated the effectiveness of these treatments first-line and in a range of patient populations, including those who are ineligible for transplantation, those with unfavourable early-stage disease, and patients older than 60 years. Therefore, anti-PD1 drugs for the treatment of HL are currently an expanding field [108].

### 5.2. Non-Hodgkin’s Lymphoma

There is high heterogeneity within NHL; some of the subtypes respond to PD-1 inhibitors, while others do not. Upregulation of PD-L1 has been observed in primary mediastinal B cell lymphoma (PMBL) [109], primary testicular lymphoma [110], plasmoblastic lymphoma, HHV-8 associated primary effusion lymphoma, and primary lymphomas of the central nervous system [111]. Pembrolizumab showed promising activity in PMBL during a phase Ib study, yielding an ORR of 41% [109]. Chronic Epstein–Barr virus (EBV) infection also plays a role in the upregulation of PD-L1 [112].

Diffuse large B cell lymphoma (DLBCL) represents a heterogeneous group of tumours derived from mature B lymphocytes. A retrospective series of 1253 patients showed that only 11% of tumours were PD-L1 positive, and these also had a poor prognosis [113]. Furthermore, patients with an International Prognostic Index (IPI) score and high PD-L1 expression have a lower prognosis, with a three-year overall survival rate of 40.9% compared to 82.1% in those with low PD-L1 expression [114]. In a phase Ib biomarker search study, the ORR for nivolumab in unselected patients with DLBCL was 36% [115]. Given these results, the evaluation of alternative biomarkers to PD-L1 in patients with NHL will permit a better selection of patients in clinical trials.

### 5.3. Multiple Myeloma

MM is a heterogeneous haematologic malignancy derived from terminally differentiated B cells known as plasma cells, which secrete monoclonal immunoglobulin paraproteins. Tumour effects in the microenvironment cause systemic multifactorial cellular and humoral immunodeficiency. The tumour microenvironment is characterized by profound immunosuppression related to several molecular pathways, including PD-1 signalling [116]. Unlike premalignant plasma cells, MM tumour cells express PD-L1 driven by IFN-gamma, toll-like receptor, Akt, and Ras signalling [117]. A retrospective study of 664 patients with MM showed that patients with a high mutational load and neoantigen expression had lower PFS rates than patients with a lower mutational load [118]. This prompted the evaluation of PD-1 inhibitor drugs for the treatment of such patients with MM and a high mutational burden.

Despite the above evidence, the use of anti-PD1 drugs in MM has not shown the expected results. In the CheckMate 039 study, no nivolumab activity was observed in 27 patients. Preclinical data supported a synergistic role of immunomodulatory drugs (IMiDs) in blocking the PD-1 pathway in patients with MM, and the preliminary results of the combination study were encouraging [115]. The phase I study of pembrolizumab with lenalidomide and dexamethasone (KEYNOTE-023) showed a response to treatment in 20/40 patients (ORR of 50%). In lenalidomide refractory patients, the combination showed an ORR of 38% [119]. Subsequently, the phase II study of pembrolizumab plus pomalidomide plus dexamethasone showed an ORR of 60%, and 8% of patients achieved a complete remission. In patients previously refractory to proteasome inhibitors such as IMiDs, the response rate was 68% [120].

Based on these results, two phase III studies of pembrolizumab in combination with lenalidomide and dexamethasone (KEYNOTE-185) or pomalidomide plus dexamethasone (KEYNOTE-183) were performed, as was a phase III study of pomalidomide plus dexamethasone versus nivolumab plus pomalidomide and nivolumab plus elotuzumab plus pomalidomide plus dexamethasone (CheckMate 602). Despite the initial optimism, in June 2017, the FDA requested that the studies be stopped due to an increase in deaths in patients in the trial arms who received pembrolizumab in combination with lenalidomide or pomalidomide [121,122]. In the interim analysis of the data from the KEYNOTE-183 study, an increased risk of death was observed in the pembrolizumab arm (HR 1.61, 95% CI = 0.91–2.85), and an ORR of 34% versus 40% was observed in the control arm [123]. In the KEYNOTE-185 study, the hazard ratio in the pembrolizumab arm was 2.06 (95% CI = 0.93–4.55), and the ORR was 64% versus 62% in the control arm [123]. These results highlight the need to explore immune regulation at the interface between malignant plasma cells and the tumour microenvironment. Further analyses and studies are expected to clarify the role of anti-PD1 and anti-PDL1 drugs in the treatment of MM.

### 5.4. Leukaemia

PD-1 pathway blocking is currently being explored as a treatment for leukaemia, often in combination with epigenetic and immunomodulatory agents. In contrast to what has been observed in lymphomas, clinical results for leukaemia have been limited. However, preclinical data suggest a potential role for the PD-1 pathway in patients with myeloid neoplasms. In AML, the expression of PD-L1 was negatively correlated with the results of the treatments [124,125,126]. Although PD-L1 expression is low in most cases of de novo AML, interferon-induced expression of PD-L1 is increased in AML blast cells during treatment, and PD-L1 expression is higher in patients with relapsed AML than in de novo patients [127].

Preclinical mouse models point to the PD-1 pathway in immune system evasion in AML. To date, preliminary clinical data for nivolumab in combination with azacitidine in patients with R/R AML also suggest promising overall response rates of up to 34%. There are several ongoing clinical trials evaluating the combination of nivolumab with other agents in precursor B-acute lymphoblastic leukaemia (B-ALL) and AML [128]. Pembrolizumab is also being evaluated in B-ALL, T acute lymphoblastic leukaemia (T-ALL), and CLL R/R [129,130].

## 6. Resistance to ICIs Secondary to MDSCs

Different populations of MDSCs are capable of mediating mechanisms of resistance to ICIs through different mechanisms. There are three fundamental ways in which MDSCs can perform this function, which are as follows [131,132]:Direct action on T cells: As the main objective of ICIs, the function of T cells is essential for the response of patients receiving treatment with ICIs, and is affected by MDSCs directly through cell–cell contact, correlating with a poor clinical outcome in patients treated with ICIs [133,134,135,136]. Some studies have shown how, in TME, the MDSCs present a high expression of PD-L1, which reduces the T cell populations in TME. After treatment with immunotherapy, the MDSCs have been observed to begin a high expression of different receptors that generate a negative regulation on the antitumour mechanisms of T cells [137,138] (Figure 1). In the future, more studies are needed to assess how the function of T cells is affected by ICIs secondary to the change in the phenotype of MDSCs.Angiogenesis secondary to MDSCs: The myeloid origin of MDSCs carries an ability to facilitate angiogenesis in TME in both solid and haematological tumours [139]. These processes have been observed to be fundamental in angiogenesis through the administration of BV8 in haematological neoplasms [140]. This specific antibody significantly reduced new blood vessels in TME, leading to a key antitumour response. Therefore, antiangiogenic drugs associated with anti-MDSC therapies could be a key point in the treatment of tumours with a high expression of MDSCs in the future [141,142]. Likewise, in the future, the combination of therapies with ICIs plus antiangiogenics and drugs that alter the number or function of MDSCs could be a new way to treat haematological neoplasms. It will be important to assess the toxicity associated with these treatments, since currently antiangiogenic drugs show partial specificity.Interaction of MDSCs with TME: Different TME cells can generate greater secondary immunosuppression on contact with MDSCs [143,144,145]. Different pathways, such as Wnt/β-catenin, can interact with MDSC receptors, generating greater tumour leakage. These interactions not only affect the immune system, but can also help angiogenesis, metalloprotease expression, or the epithelium–mesenchyme transition. There are currently various drugs in clinical trials that are directed against the microenvironment in different neoplasms. These drugs will be able to modify the phenotype of the MDSCs and their associated immunosuppression. The combination of these therapies with immunotherapy may make it possible to enhance the activating effect of the immune system through the direct activation of T cells by ICIs, as well as the decrease and alteration of the function of MDSCs.

In summary, the three key mechanisms that lead to immunotherapy resistance (ICIs) secondary to MDSCs are: (i) the interactions of T cells with MDSCs, (ii) the angiogenesis generated by the populations of MDSCs, and (iii) the interactions between TME and MDSCs that lead to a more immunosuppressive environment. The combination of these different strategies will enable the optimization of anti-MDSCs treatments.

## 7. Targeting MDSCs to Overcome Resistance to ICIs

Higher levels of MDSCs are observed in patients with haematologic cancer and in those with a more advanced or relapsed stage. A study conducted at DLBCL showed that PD-L1 expression is key in MDSC-induced T cell suppression [146,147]. Likewise, this has also been seen in high-risk Sokal patients with CML and with high levels of MDSCs; this is accompanied by an increase in the expression of PD-L1 [148]. These studies suggest that antitumour therapies currently under development that involve the combination of ICIs and targeting MDSCs may be effective in the future for the treatment of haematological malignancies.

Recent preclinical studies have shown how immunotherapy treatments associated with therapies directed at MDSCs are capable of producing a greater antitumour effect than that generated by ICIs in monotherapy. The immunomodulatory response generated after the administration of ICIs can be disrupted by targeting the MDSCs, which presents a unique opportunity for the treatment of haematological malignancies [149].

### 7.1. ICIs and Reduction of the Number of MDSCs

Under pathological conditions, the number of MDSCs increases significantly. A two-phase model that describes this process has been proposed by Condamine and Gabrilovich. The first phase contains an expansion of immature myeloid cells correlated with the block of terminal differentiation of haematopoietic stem cells, and the second phase involves the activation of immature myeloid cells to MDSCs [150]. Although the dominant factors of each phase overlap significantly, it has been observed that there is a predominance of tumour-derived growth factors in the first phase (GM-CSF and G-CSF) and later, in the second cytokine phase, of pro-inflammatory diseases derived from the tumour stroma (IL-1b or IL-6) [151]. Thus, reducing the frequency of MDSCs can help to end myelopoiesis normally.

Some chemotherapy drugs have shown effects on the number of MDSCs in tumour carriers, especially at a solid level. Gemcitabine (a nucleoside analogue) lowers the level of splenic MDSCs. In mesothelioma-derived studies in mice, gemcitabine in synergy with ICIs was observed to be more effective than gemcitabine or ICIs alone [152]. In small-cell lung cancer, the addition of SRA737 (an oral CHK1 inhibitor) to gemcitabine and ICIs increases the efficacy of the treatment, being associated with a greater decrease in MDSCs and Treg cells [153]. Likewise, regarding 5-FU (a promoter of IL-1b secretion by MDSCs), it has been observed that, in mice bearing colorectal cancer xenografts, there is increased survival associated with 5-FU and anti-PD1 combined treatment, compared to 5-FU in monotherapy [154,155]. These observations have been reported not only for chemotherapy, but also for tyrosine kinase inhibitors, such as dasatinib. In head and neck models, dasatinib treatment facilitated anti-CTLA4 immunotherapy by decreasing the population of MDSCs and increasing the proportion of CD8 cells [156]. A clinical trial is currently underway (NCT03516279) to evaluate the efficacy of pembrolizumab plus dasatinib, imatinib mesylate, or nilotinib in patients with CML and a persistently detectable minimal residual disease. The phase Ib trial NCT02011945 has already evaluated the efficacy of a double blockade with dasatinib plus nivolumab in patients with CML; however, this study did not show the expected results, and lacked preliminary efficacy data in terms of survival [157].

T cell large granular lymphocytic leukaemia is a tumour that mainly expresses late inflammatory chemokine receptors, such as CXCR1 and CXCR2. In preclinical models, CXCR2 antagonists have been shown to be potential inhibitors of the recruitment of MDSCs, especially PMN-MDSCs [158]. The combination of anti-PD1 therapy together with anti-CXCR2 is a potent therapeutic target for the treatment of these leukaemias. The double blockade prevented the trafficking of MDSCs and restored the antitumour effects of the delayed treatment with ICIs.

### 7.2. ICIs and Functional Alteration of MDSCs

Entinostat (histone deacetylase inhibitor) eradicates 80% of tumour cells and reduces MDSCs in combination with anti-CTLA4 and anti-PD1 antibodies, where monotherapy with ICIs has not achieved responses [159]. A similar result has been obtained with other inhibitors, such as mocetinostat, in combination with anti-PDL1 antibodies [160]. The theory by which synergy is valued has been based on the fact that the reduction of ARG1, iNOS, and COX-2 in the MDSCs increases the efficacy of the inhibition of the PD-1 pathway [161]. In this regard, a clinical trial (NCT02936752) is currently evaluating the efficacy of pembrolizumab plus entinostat in patients with myelodysplastic syndrome after unsuccessful treatment with DNA methyltransferase inhibitors (DNMTi) [144].

Another drug that has shown significant activity in altering the function of MDSCs is ibrutinib, which targets Bruton’s tyrosine kinase (BTK) and is widely used for the clinical treatment of B cell neoplasms. This drug has been reported to be capable of attenuating nitrate reductase (NO) production and indoleamine 2,3-dioxygenase (IDO) expression from MDSCs, improving the efficacy of anti-PDL1 therapies [162]. Along with the above, IDO inhibitors also appear to have shown significant preclinical activity in combination with ICIs [163]. The clinical trial NCT02332980 is currently evaluating pembrolizumab monotherapy versus a combination with idelalisib or ibrutinib in patients with CLL R/R or other low-grade B cell NHL [164].

Trans-retinoic acid (ATRA) is a derivative of vitamin A that induces the differentiation of immature myelocytic cells in patients with APL. Likewise, ATRA acts on MDSCs to promote their differentiation, and leads to a decrease in the frequency of MDSC circulation through the activation of ERK1/2, the regulation of glutathione synthase, and the generation of glutathione [164]. Studies in metastatic melanoma have shown an increase in survival for the combination of anti-CTLA4 with ATRA, which could open a way for the treatment of different leukaemias in the future [165,166]. We have collected the main clinical trials for the MDSC and ICI combination in Table 3.

## 8. Future Paths in MDSCs and ICIs

As a highly heterogeneous cell, MDSCs present considerable difficulty in conducting clinical studies due to the lack of a standardized method in differentiating normal neutrophils from MDSCs [167]. Future research involves first determining unique or selective biomarkers of these cells [168,169], as well as understanding the regulations and pathways of ETM on the effect of MDSCs and their interaction with Treg cells [170]. Despite the research needs in this field, the role of MDSCs in haematological neoplasms is already widely known, which is related to more aggressive tumours and lower survival rates than to tumours with low levels of MDSCs [171]. Given the above and the important role that ICIs have in the treatment of haematological neoplasms, especially at the level of lymphomas, various clinical trials are being evaluated in this double blockade.

A range of drugs are being evaluated for correcting the number or function of MDSCs in haematological tumours. Recently, Lewinsky et al. [172] showed how the negative regulation or blocking of CD84 in multiple myeloma allows for decreasing the subpopulations of MDSCs with the consequent activation of T cells.

The double blocking of MDSCs and ICIs should take into account possible interactions, because ICIs may have an unknown regulation, both positive and negative, on the synthesis or function of MDSCs. Also important to consider in the role of the combination of these therapies are the unknown relationships between graft versus leukaemia, graft versus host disease, and MDSCs [173,174]. In this area, more preclinical studies are needed that can be translated into clinical trials on the combination of MDSC modification therapies with ICIs, since a greater understanding of the role of MDSCs in haematological neoplasms is necessary.

## 9. Conclusions

Currently, ICIs present significant limitations in the treatment of haematological neoplasms, although there are already established results that are very promising for the treatment of lymphomas. The limited benefit in patients with other haematologic malignancies is a major obstacle associated with ICI therapies. Further research is needed to pave the way for these treatments, in which MDSCs may have possible responses to the current lack of a clinical benefit. The combination of double blockade with drugs that function on MDSCs plus ICIs may come to play a key role in the treatment of different haematological neoplasms. Further preclinical and clinical studies will be required to elucidate the true role of double blockade.

## Figures and Tables

**Figure 1 jcm-10-01919-f001:**
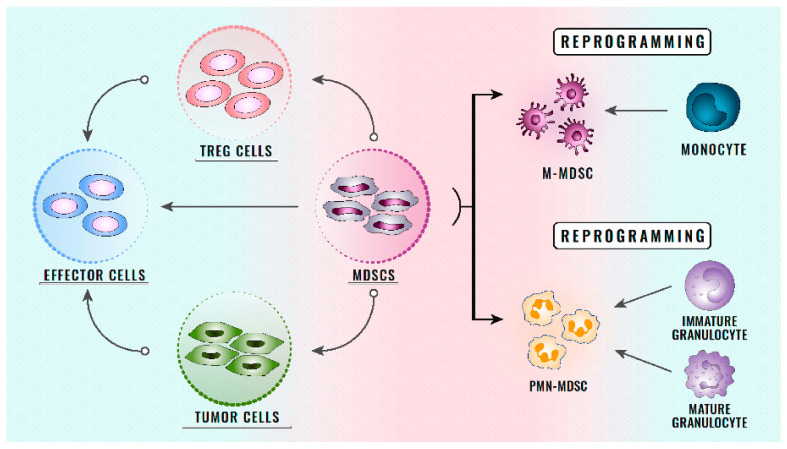
Mechanisms of immune suppression of MDSCs on T cells and tumour cells. This figure illustrates the origin of the MDSC populations. After the change in the phenotype of myeloid cells, they present different interactions with Treg cells and tumour cells that lead to dysregulation of the immune system. Alterations of the immune system lead to an increase in immunosuppression, with a consequent increase in cell proliferation and immune resistance mechanisms.

**Table 1 jcm-10-01919-t001:** Differential expression markers in the populations of MDSCs in animal models.

MDSC Population	Chemokine Receptor	Cluster of Differentiation *	MHC	Ly6	Mac-2 (Galectin-3)
**M-MDSC**	CCR2 (high) CCR5 CX3CR1 CXCR1 CXCR2 CXCR4	CD1b CD49d CD54 (high) CD71 CD73	Class I Class II (+/−)	Ly6C + (high)	High
**PMN-MDSC**	CCR2 CCR5 CX3CR1 CXCR1 CXCR2 CXCR4	CD43 CD54 (low) CD73 (high) CD98 CD244 (+/−)	Class I Class II (+/−)	Ly6C + (low) Ly6G + (high)	Low

* This table shows the different markers that are used in animal models for the identification of MDSC populations. The main markers of differentiation are the cluster of differentiation. The expression of galectin-3 also allows clearer differentiation than other markers, such as Ly6 or MHC.

**Table 2 jcm-10-01919-t002:** Differential expression markers in the clinical identification of MDSC populations.

Type of MDSC	Chemokine Receptor	Cluster of Differentiation *	HLA-DR	Lin
**M-MDSC**	CCR2 (high) CXCR4 CXCR1 CXCR2	CD14 (high) CD68 CD80 (+/−) CD83 (+/−) CD86 (+/−) CD163 CD117 (+/−)	Negative	+/− (low)
**PMN-MDSC**	CCR2 CXCR4 CXCR1 CXCR2	CD15 CD66b CD117	Negative	+/− (low)

* This table shows the different markers that are used in the clinic for the identification of MDSC populations. There are a series of markers that are jointly expressed in M-MDSC and PMN-MDSC, which do not allow differentiating the populations. The main markers of differentiation are the cluster of differentiation. Neither M-MDSC nor PMN-MDSC show HLA-DR or Lin expression.

**Table 3 jcm-10-01919-t003:** Main clinical trials underway in haematological malignancies of combined strategies against MDSCs and ICIs.

Haematological Malignancies	NCT Number	Interventions
CML, BCR-ABL1-positive minimal residual disease	NCT03516279	Pembrolizumab and dasatinib, imatinib mesylate, or nilotinib in treating patients with CML and persistently detectable minimal residual disease
Myelodysplastic syndrome	NCT02936752	Entinostat and pembrolizumab in treating patients with myelodysplastic syndrome after DNMTi therapy failure
Recurrent CCL	NCT02332980	Pembrolizumab alone or with idelalisib or ibrutinib in treating patients with R/R chronic lymphocytic leukaemia or other low-grade B cell NHL

This table shows the main clinical trials in haematological neoplasms that assess the action of anti-MDSCs drugs together with ICIs. The main immunotherapy drug used is pembrolizumab, which is combined with drugs of different action against MDSCs.

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
