# Peer review of "Resistance to Immune Checkpoint Inhibitors Secondary to Myeloid-Derived Suppressor Cells: A New Therapeutic Targeting of Haematological Malignancies"

_jcm, 2021, doi:10.3390/jcm10091919_

Round 1
Reviewer 1 Report
The review entitled "Resistance to immune checkpoint inhibitors secondary to myeloid-derived suppressor cells: a new therapeutic targeting of hematological malignancies", is a review that focuses on MDSC and ICIs and the potential clinical implications in hematological malignancies. The comprehensive summary of these studies is definitely of interest. However, the manuscript could be significantly improved.
General comments:
- Parts: "MDSCs and hematological malignancies" and "Immunotherapy for hematological malignancies" - do not contribute anything new to the field of study review is focused on. In the bibliography it is possible to find numerous reviews with the same content of these paragraphs, some even used by the authors in their references. Lv and Wang et al. (J Hematol Oncol. 2019 Oct 22;12(1):105. doi: 10.1186/s13045-019-0797-3) reviewed the MDSCs in hematological cancers of frequency, characters and mechanisms. Salik et al (J Hematol Oncol. 2020 Aug 12;13(1):111. doi: 10.1186/s13045-020-00947-6) discuss recent advances and emerging roles of immune checkpoint blockade in hematological malignancies.
- In my opinion, authors should focus more on the sections: "Resistance to ICIs secondary to MDSCs" and "Targeting MDSCs to overcome resistance to ICIs". I think it would be useful to expand these parts of the review. However, some of the data reported here have been already described by other groups. Hou et al. (Front Immunol. 2020;11:783. doi:10.3389/fimmu.2020.00783) discussed the role of MDSCs in resistance to ICIs and summarized the therapeutic strategies targeting them to enhance ICIs efficiency in cancer patients.
- Tables 1 and 2 do not contribute anything new. Likewise, Figure 1 is not attractive and does not contribute anything new.
- Tables and figures should be self-contained. The figure/tables legends should be detailed enough that readers can understand the details without reading the main text. The legends should be improved.
- The authors should also make sure that the numerous abbreviations are systematically explained as many were not.
Specific comments
- Población (Table 2) is not a term used in English
- Lines 102-110: for such a broad claim, some references should be added.
- Line 114: reference [3,4] should be checked.
- Lines 114-117: add a reference for this statement
- Line 281: Please check the reference. It should probably be 96.
- What do the abbreviations LH or Wnf mean?
- References: 26,27,28, 34 and 143 are not referenced in the main text.
Reviewer 2 Report
The review of Olivares-Hernández et al. provides interesting insights in the role of MDSCs conferring resistance to immune checkpoint inhibitors in hematologic malignancies. The topic is interesting and the review is innovative. Overall the review is well and clearly written.
I have only some minor issues:
The paragraph leukemias should be divided in acute leukemias, myeloproliferative neoplasms and CLL.
Furthermore, the potential role of MDSCs in the pathophysiology of myelodysplastic syndromes and maybe their implication in resistance to ICIs should be included in the manuscript.
Reviewer 3 Report
This review manuscript is highlighting the resistance to immune checkpoint inhibitors secondary to myeloid-derived suppressor cells (MDSCs) with a perspective of a new therapeutic targeting of hematological malignancies. Overall, this is a well-written and detailed manuscript with literature research. However, the authors missed including some very recent publications (References 1,2) related to the role of MDSCs in hematological malignancies. I would agree that the authors did extensive research if those publications would be reviewed.
Reference 1-Gunes, EG et al. “The role of myeloid-derived suppressor cells in hematologic malignancies.” Current Opinion in Oncology: September 2020 - Volume 32 - Issue 5 - p 518-526 doi: 10.1097/CCO.0000000000000662
Reference 2-Lewinsky, Hadas et al. “CD84 is a regulator of the immunosuppressive microenvironment in multiple myeloma.” JCI insight vol. 6,4 e141683. 22 Feb. 2021, doi:10.1172/jci.insight.141683
The introduction part is diffused and short of a central hypothesis and focus. The transition to ICIs is not clear.
In the definition and role of MDSCs part, there are vague sentences; it was difficult to understand. The interpretation of the chronic inflammation in cancer is most strongly induced by MDSCs in the tumor microenvironment is unclear. There is a discussion about BMI spectrum; what is that about? Was that a word processing error?
The expression markers in the clinical identification of MDSCs populations are questionable, as the legends do not match the markers of the MDSCs used in the tables.
Further, the authors would adequately document how suppressive function of MDSCs in TME, changes in MDSC phenotypes could occur with ICIs therapies, how alterations in MDSC expression of inhibitory molecules affect T cell functions in hematological malignancies.
In several cases, the words are written inaccurately.
Overall, the manuscript about the resistance to immune checkpoint inhibitors and MDSCs with a perspective of a new therapeutic targeting of hematological malignancies is not ready to publish.
Round 2
Reviewer 1 Report
The authors have replied to most of the comments made in the previous review. The manuscript appears substantially more complete and its present form does not raise my objections.
This manuscript is a resubmission of an earlier submission. The following is a list of the peer review reports and author responses from that submission.